# Select the right agent: Data-Driven Online Model Selection in Reinforcement Learning

## Abstract

We study the problem of online model selection in reinforcement learning, where the selector has access to a class of reinforcement learning agents and learns to adaptively select the agent with the right configuration. Our goal is to establish the improved efficiency and performance gains achieved by integrating online model selection methods into reinforcement learning training procedures. We examine the theoretical characterizations that are effective for identifying the right configuration in practice, and address three practical criteria from a theoretical perspective: 1) Efficient resource allocation, 2) Stabilized training, 3) Adaptation under non-stationary dynamics. Our theoretical results are accompanied by empirical evidence from various model selection tasks in reinforcement learning, including neural architecture selection, step-size selection, and self-model selection.

## 1 Introduction

A major effort in theoretical analysis of reinforcement learning algorithms is towards providing guarantees on the regret or sample complexity (Agarwal et al., 2019; Foster and Rakhlin, 2023). These guarantees often rely on assumptions and problem-specific constant that are left to be configured at deployment. As a result, one would expect the theoretical guarantee to hold in practice under the assumption that the agent is well-specified; the configuration of the algorithm is consistent with the true nature of the problem instance. Otherwise, in case of what is known as misspecification Joseph et al. (2013); Li and Yang (2024), we observe a mismatch between the theoretical guarantee and the realized performance of the agent.

Model Selection offers a remedy to the theory-practice mismatch that arises from misspecification. Given access to a set of base agents, the model selection algorithm, which we will refer to as the selector, interacts with an environment and learns to select the right model for the problem at hand. The algorithmic goal of model selection is to guarantee that the meta-learning algorithm for the selector has performance comparable to the best *solo* base agent, without knowing a priori which agent is best suited for the problem at hand.

While theoretical model selection guarantees have been established across various sequential decision-making problems (Pacchiano Camacho, 2021; Cutkosky et al., 2021; Lee et al., 2022), this work focuses on online Reinforcement Learning (RL). We propose a training mechanism that integrates online model selection methods into RL training procedure with minimal intervention. The value of online model selection in RL lies in identifying the right configuration for the agent on the fly and using the reward feedback to spend the compute budget for training more effectively. We analyze the performance of our training mechanism across three model selection tasks and establish corresponding theoretical properties for data-driven online model selection:

1. We derive the relationship between the realized performance of each base agent and the compute that is allocated to them by the selector. We show that for model selection algorithms that satisfy a balancing property, the selector learns to direct more compute to base agents with better realized performance.

2. Under non-stationary dynamics where the optimal base agent is likely to change, data-driven model selection can adapt to the new optimal choice.

3. We prove a simple argument showing that in scenarios where the performance of the RL algorithm is sensitive to the randomness of the initialization, data-driven online model selection can stabilize the training by leveraging multiple initializations of the algorithm within a single run.

We leverage model selection methods that neither require structural assumptions about the RL agents nor their theoretical regret bounds. These algorithms use the realized performance of agents to do model selection in a data-driven manner Dann et al. (2024). Thus, our training mechanism is algorithm-agnostic and can be applied to any RL algorithm. To validate these properties, we conduct experiments on neural architecture selection and step-size selection for different deep RL algorithms. Additionally, we evaluate various model selection algorithms within our framework, highlighting the significance of data-driven selection. Our work has close connections to meta learning algorithms for adaptive hyperparameter selection in RL Elfwing et al. (2017); Parker-Holder et al. (2022), and addresses a key challenge of continual learning in non-stationary environments. [1]

## 2 PRELIMINARIES

### 2.1 REINFORCEMENT LEARNING

We consider episodic reinforcement learning with a finite horizon $H$, which is formalized as a Markov Decision Process (MDP) $\langle \mathcal{S}, \mathcal{A}, \mathcal{R}, \mathcal{P}, \rho \rangle$. Here, $\mathcal{S}$ denotes the state space, $\mathcal{A}$ is the action space, $\mathcal{R} : \mathcal{S} \times \mathcal{A} \to \mathbb{R}$ is the reward function, $\mathcal{P} : \mathcal{S} \times \mathcal{A} \to [0, 1]$ is the environment transition probabilities, and lastly $\rho : \mathcal{S} \to [0, 1]$ is the initial state distribution. Denote $\pi : \mathcal{S} \to \Delta(\mathcal{A})$ as the policy of the agent. The agent interacts with the MDP according to the following procedure: At each step $h \in [H]$, the agent takes action $a_h \sim \pi(s_h)$, observes $r_h = \mathcal{R}(s_h, a_h)$ and move to state $s_{h+1} \sim \mathcal{P}(s_h, a_h)$. The value of the policy is defined as,

$$v(\pi) = \mathbb{E}\left[\sum_{h=1}^{H} r_h\right] \tag{1}$$

Here, the expectation is with respect to the stochasticity of the interaction procedure. The goal of the agent is to learn an optimal policy defined as,

$$\pi^* = \arg\max_{\pi \in \Pi} v(\pi) \tag{2}$$

where $\Pi$ denotes the policy class. The policy class is commonly parameterized as $\Pi = \{\pi_\theta : \theta \in \Theta\}$, where $\pi^* = \pi(\theta^*)$. The state value function $V : S \to \mathbb{R}$ and state-action value function $Q : S \times A \to \mathbb{R}$ with respect to policy $\pi$ are defined as,

$$V^\pi(s) = \mathbb{E}\left[\sum_{h=1}^{H} r_h | s_0 = s\right] \tag{3}$$

$$Q^\pi(s, a) = \mathbb{E}\left[\sum_{h=1}^{H} r_h | s_0 = s, a_0 = a\right] \tag{4}$$

Note that definition 1 of the value function is equivalent to $v(\pi) = \mathbb{E}_{s \sim \rho}[V^\pi(s)]$, and $v(\pi) = \mathbb{E}_{s \sim \rho, a \sim \pi}[Q^\pi(s, a)]$. From the algorithmic perspective, the agent does not try to directly solve the optimization problem in 2, as it can be computationally expensive, and in certain cases intractable. Instead, RL algorithm iteratively update the parameter $\theta$ to optimize the state-value function $V^\pi(s)$ or the state-action value function $Q^\pi(s, a)$. Much of RL is devoted to designing algorithms with effective and sample-efficient update rules. But the most prominent themes are optimizing the advantage function, $A^\pi(s, a) = Q^\pi(s, a) - V^\pi(s, a)$ through the policy gradient method (Williams, 1992; Schulman et al., 2017), or optimizing the state-action value function by temporal difference methods (Watkins and Dayan, 1992; Mnih et al., 2015), which we include in Appendix C.2. The details of the RL algorithm is not the focus this work, as we will show later that online model selection methods can be integrated into any RL training procedure.

---

[1]Our codes are open-sourced and uploaded as supplementary material.

**Algorithm 1** Selector (D³RB)

---

**Input:** $M, \mathcal{B}, \hat{d}_0^i = d_{min} \; \forall i \in [M]$
**Function** `sample()`:
    $i = \arg\min_j \phi_t^j$
    **return** $\mathcal{B}^i$
**Function** `update`$(i, r, t)$:
    $u_{t+1}^i = u_t^i + r, \qquad n_{t+1}^i = n_t^i + 1$
    Perform Misspecification test 12 for $\mathcal{B}^i$
    **if** *Test Triggered* **then**
        $\hat{d}_t^i \leftarrow 2\hat{d}_t^i$
        $\phi_t^i = \hat{d}_t^i \sqrt{n_t^i}$

**Algorithm 2** Selector+RL Training Mechanism

---

**Input:** Model Selector $selector$, $T$, $H$
**for** $t = 1, 2, ..., T$ **do**
    $i_t = selector.\text{sample}()$
    **for** $h = 1, 2, ..., H$ **do**
        $a_h \sim \pi_t^{i_t}(\cdot | s_h)$
        $r_h \leftarrow \mathcal{R}(s_h, a_h)$
        $s_{h+1} \leftarrow \mathcal{P}(s_h, a_h)$
    Forward $\{(s_h, a_h, r_h)\}_{h=1}^H$ to base agent $\mathcal{B}^{i_t}$
    to update it policy
    $R_t = \sum_{h=1}^H r_h$
    $selector.\text{update}(i_t, R_t, t)$

Figure 1: Online Model Selection Framework in RL

## 2.2 Model Selection

We consider the online model selection problem where the selector has access to a set of $M$ base agents,

$$\mathcal{B} = \{\mathcal{B}^1, \ldots, \mathcal{B}^M\}$$

At round $t \in [T]$, the selector picks base agent $i_t \in [M]$ according to its selection strategy. The selector rolls out the policy of the base agent $\pi_t^{i_t}$ for one episode and collects the trajectory,

$$\tau = \{(s_h, a_h, r_h)\}_{h=1}^H$$

according to the procedure explained in 2.1. The selector then uses this trajectory to update its selection policy, and forwards it to the base agents so that it can update its internal policy $\pi_t^i$.

**Notation:** Denote $n_t^i = \sum_{l=1}^t \mathbb{I}[i_l = i]$, as the number of rounds that base agent $\mathcal{B}^i$ has been selected up to round $t \in [T]$. Denote, $u_t^i = \sum_{l=1}^t \mathbb{I}[i_l = i] r_l$, where $r_l$ is the episodic reward at round $l \in [T]$, and $\bar{u}_t^i = \sum_{l=1}^t \mathbb{I}[i_l = i] \mathbb{E}[r_l \mid \pi_l^i]$. The regret of base agent $i \in [M]$ up to time $t \in [T]$,

$$\text{Regret}_t^i = n_t^i v^* - \bar{u}_t^i \tag{5}$$

where $v^* = max_{\pi \in \Pi} v(\pi)$. The total regret of the selector after $T$ rounds is,

$$\text{Regret}(T) = \sum_{i=1}^M \text{Regret}_T^i = \sum_{i=1}^M n_T^i v^* - \bar{u}_T^i \tag{6}$$

The selector has access to base agents as sub-routines, and sequentially picks them in a meta-learning structure.

## 3 Training Mechanism

Online Model Selection in Reinforcement Learning deals with the problem of selecting over a set of *evolving* policies. In this section, we characterize how the selector keeps track of the performance of base agents over time, and what is the right measure for that purpose. Importantly, we use this measure to specify the algorithmic objective of model selection.

**Definition 1** (Regret Coefficient). *For a positive constant $d_{min}$, denote the regret coefficient of base agent $\mathcal{B}^i \in \mathcal{B}$ at time $t \in [T]$ as,*

$$d_t^i = \max\left(\frac{\text{Regret}_t^i}{\sqrt{n_t^i}}, d_{min}\right) \tag{7}$$

**Algorithmic Objective in Model Selection** The goal in Model Selection is to design a meta-algorithm for the selector that has comparable performance to the base agent with minimum realized regret at the end of round $t \in [T]$. Define,

$$d_* = \min_{i \in [M]} \max_{t \in [T]} d_t^i \tag{8}$$

as the regret coefficient of the well-specified base agent. The total regret incurred by the meta-algorithm should satisfy,

$$\text{Regret}(T) = \sum_{i \in [M]} \text{Regret}_T^i \leq \text{Poly}(d_*)\sqrt{T} \tag{9}$$

with high probability.

### 3.1 ALGORITHM

Algorithm 1-left shows the Doubling Data Driven Regret Balancing algorithm (D³RB) (Dann et al., 2024) that is the selector of interest in this paper. Algorithm 1-right is the training mechanism, that shows how to integrate a selector into the RL training loop. The training mechanism is analogous to the standard agent-environment interaction protocol in RL with minimum interventions from the selector. As a result, this framework can be integrated into any RL algorithm. The choice of selector is also not limited to D³RB, and other model selection algorithm can be used as long as they follow a similar interface to Algorithm 1-left. With that being said, the choice of selector matters in the performance and efficiency of the training mechanism. We choose D³RB that satisfies the following high probability model selection guarantee,

**Theorem 2** ((Dann et al., 2024)). *Denote the event $\mathcal{E}$,*

$$\mathcal{E} = \left\{ \mid u_t^i - \bar{u}_t^i \mid \leq c\sqrt{n_t^i \ln \frac{M \ln n_t^i}{\delta}} \right\} \tag{10}$$

*for a parameter $\delta$ and universal constant $c$. Under event $\mathcal{E}$, the total regret incurred by data-driven regret balancing (D³RB) after $T$ rounds satisfies,*

$$\text{Regret}(T) = \mathcal{O}\left(d_* M \sqrt{T} + d_*^2 \sqrt{MT}\right) \tag{11}$$

*With probability $1 - \delta$.*

This bound implies that D³RB is optimal as it matches the lower bound of online model selection in (Pacchiano et al., 2020c; Marinov and Zimmert, 2021).

D³RB keeps track of a balancing potential function $\phi_t^i = \hat{d}_t^i \sqrt{n_t^i}$ for each base agent $i \in [M]$. This potential function estimates a *data-adaptive* upper bound on the realized regret. Here, $\hat{d}_t^i$ is an active estimate of the true regret coefficient $d_t^i$. At each round $t \in [T]$, the selector picks the base agent with minimum balancing potential $i_t = \arg\min_{i \in [M]} \phi_t^i$. It acts according to the policy of the selected base agent $\pi_t^{i_t}$ for one episode, collects the trajectory $\tau = \{(a_h, s_h, r_h)\}_{h=1}^H$, and updates policy $\pi_t^{i_t}$ according to the update rule of the RL algorithm. The selector updates the statistics of the selected base agent, $n_t^{i_t}$, and $u_t^{i_t}$, and doubles $\hat{d}_t^{i_t}$ if the agent is misspecified.

**Proposition 3** (Misspecification Test). *Base agent $i \in [M]$ is misspecified if it triggers the following test,*

$$\frac{u_t^i}{n_t^i} + c\sqrt{\ln \frac{M \ln n_t^i}{\delta}} + \frac{\hat{d}_t^i \sqrt{n_t^i}}{n_t^i} \leq \max_{j \in [M]} \frac{u_t^j}{n_t^j} - c\sqrt{\ln \frac{M \ln n_t^j}{\delta}} \tag{12}$$

We provide a detailed explanation in appendix A.1 on why this test can determine misspecification of a base agent. In the following sections, we analyze the properties of the D³RB+RL training mechanism in theory and demonstrate the effectiveness in experiments.

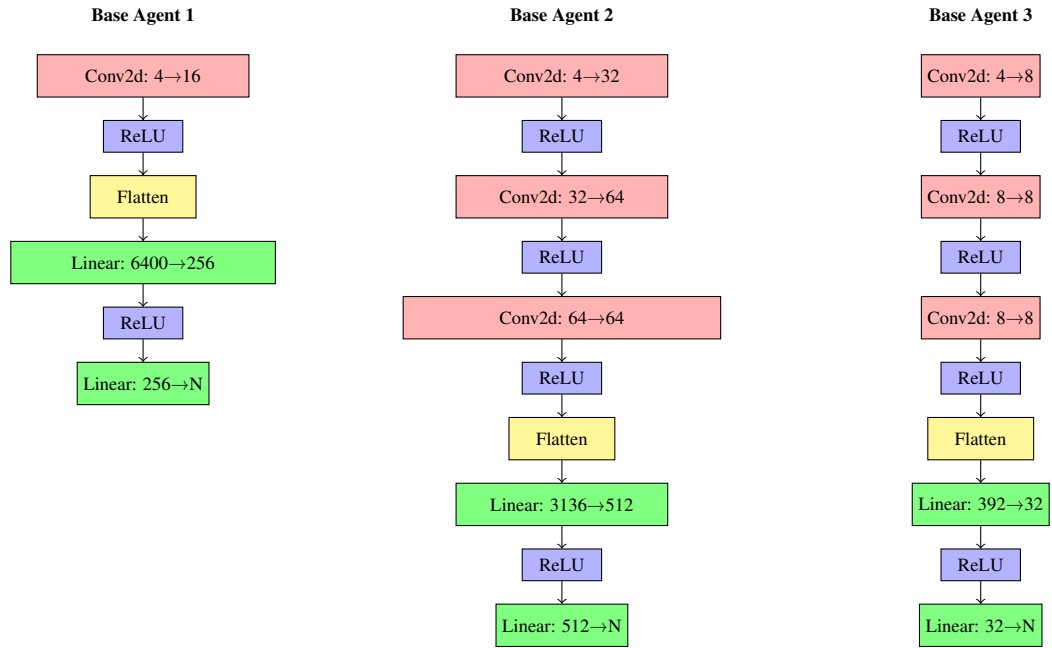

Figure 2: Q-Network architecture for three base agents in neural architecture selection task

## 4 MODEL SELECTION TASKS

### 4.1 NEURAL ARCHITECTURE SELECTION

The policy in deep RL is parametrized by a neural network, which serves as an expressive function approximator, enabling decision making in high-dimensional state spaces that was otherwise not possible by linear function approximation or tabular RL. The choice of network architecture highly affects the agent's performance as it determines the representational capacity of the agent.

To validate the practical efficiency of our mechanism, we run experiments for the neural architecture selection task in DQN agents for 4 different Atari environments Mnih et al. (2015). We pair the $D^3$RB selector with three base agents that are identified with their unique Q-network architecture. Figure 2 depicts the architecture of the Q-network for each base agent. The results of the neural architecture selection task are shown in Figure 3. To better see the progressive performance of the selector, we also included the independent execution of base agents. We observe that the performance of base agents varies drastically depending on the architectural choice, as base agent $\mathcal{B}^3$ is failing in almost all of the environments, base agent $\mathcal{B}^1$ has sub-optimal performance, and base agent $\mathcal{B}^2$ is the oracle-best. Importantly, the reward curve of the model selection approach shows that the $D^3$RB selector is able to reach the performance of the oracle-best base agent, confirming the theoretical objective of model selection 9 in practice. In the following theorem, we derive the relationship between the allocated compute and regret coefficient of each base learner under Data-Driven Model Selection.

**Theorem 4.** *Denote $\alpha_t^i$ as the fraction of time that base agent $\mathcal{B}^i$ hads been selected up to round $t \in [T]$.*

$$\alpha_t^i = \frac{\sum_{l=1}^t \mathbb{I}[i_l = i]}{t}, \qquad \forall i \in [M], t \in [T]$$

*Then, Data-driven model selection ($D^3$RB) satisfies,*

$$\alpha_t^i = \frac{(1/d_t^i)^2}{\sum_{j=1}^M (1/d_t^j)^2}$$

*where $d_t^i$ is the regret coefficient 1 of base agent $\mathcal{B}^i$ at time $t \in [T]$.*

*Proof.* Appendix A.2

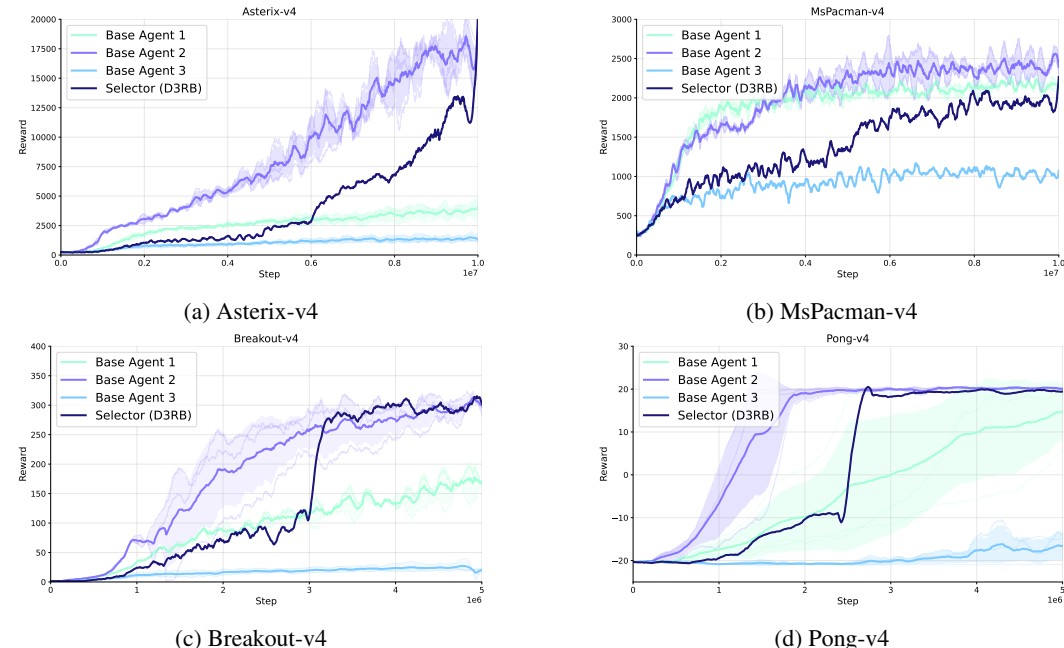

(a) Asterix-v4                       (b) MsPacman-v4

(c) Breakout-v4                      (d) Pong-v4

Figure 3: Neural Architecture Selection for DQN algorithm in Atari Environments. Comparison of D³RB with individual runs of base agents shows that the selector has comparable performance to the best solo base agent. Curves show the average and standard deviation over three seeds.

Theorem 4 shows that for a fixed training budget $T$, the base agent with a smaller regret coefficient (and hence a better realized performance) will receive a bigger fraction of compute under model selection with D³RB. Table 1 summarizes the resource allocation of 3. For each environment, we calculated the fraction of rounds in which a base agent was selected throughout training. These results show that D³RB has learned to allocate more compute to the oracle-best base agent ($\mathcal{B}^2$).

| Environment | $\mathcal{B}^1$ | $\mathcal{B}^2$ (Oracle-Best) | $\mathcal{B}^3$ |
|---|---|---|---|
| Breakout-v4 | 0.29 | **0.43** | 0.28 |
| MsPacman-v4 | 0.40 | **0.41** | 0.19 |
| Asterix-v4 | 0.29 | **0.46** | 0.25 |
| Pong-v4 | 0.21 | **0.58** | 0.21 |

Table 1: Resource Allocation of D³RB. Each number shows the fraction of time that D³RB has selected each base agent during training. The maximum amount of selection is in bold. Results reflect that D³RB selector learns to direct more compute towards the oracle-best base agent, without knowing that a priori.

.

## 4.2 STEP-SIZE SELECTION

In this section, we consider the step size selection task in RL, where we are interested in adaptively selecting the right step-size for a given RL problem. We use this task to answer why bandit algorithms are insufficient to perform model selection for RL agents and compare data-driven model selection methods with other baselines.

### 4.2.1 BANDIT ALGORITHMS FOR ONLINE MODEL SELECTION: WHAT WOULD GO WRONG?

As an algorithmic counterpart to online model selection, consider the problem of Multi-Armed Bandits (MAB)(Lattimore and Szepesvári, 2020) where the learner interacts with a finite set of arms. One can consider using a MAB algorithm for the model selection problem, where each arm index

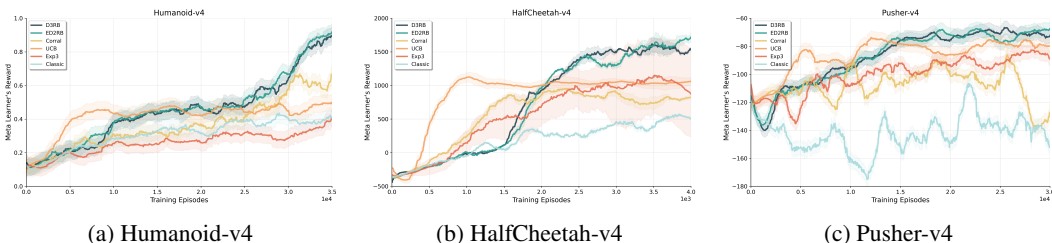

(a) Humanoid-v4          (b) HalfCheetah-v4          (c) Pusher-v4

Figure 4: Comparison of 6 model selection strategies in the step-size selection task for the PPO algorithm. Each curve shows the average and standard deviation over three seeds.

corresponds to a base agent and the MAB algorithm acts as a selector. We claim that standard Bandit algorithms with regret-minimization objective are not always suitable to perform model selection in RL.

1. The standard assumption in MAB is that each arm has an unknown but fixed reward distribution, whereas in model selection for RL agents, $\mathbb{E}[r \mid \pi_t^i]$ changes as policy gets updated over time.

2. The optimal arm in standard MAB is associated with a fixed arm index. In model selection, the optimal base agent can change in different stages of training. As an example, consider two base agents that one performs worse than the other initially, but outperforms towards the end of training. Ideally, the model selection strategy should take this non-stationarity into account and adapt to the new optimal.

The results in the figure 4 empirically validate the above. In these experiments, we consider the step size selection for PPO agents in three different MuJoco environments (Tassa et al., 2012). We initiate five PPO base agents with the celebrated logarithmic scale values for hyperparameter tuning $[1e^{-2}, 1e^{-3}, 1e^{-4}, 1e^{-5}, 1e^{-6}]$. We pair these base agents with six different model selection algorithms. The model selection strategies are $D^3RB$, $ED^2EB$ (Dann et al., 2024), Corral (Agarwal et al., 2017), Regret Bound Balancing Pacchiano et al. (2020b), which we will refer to as Classic Balancing. We consider two standard Bandit algorithms, Upper Confidence Bound (UCB) (Auer et al., 2002), and Exponential-weight algorithm for Exploration and Exploitation (EXP3) (Bubeck et al., 2012).

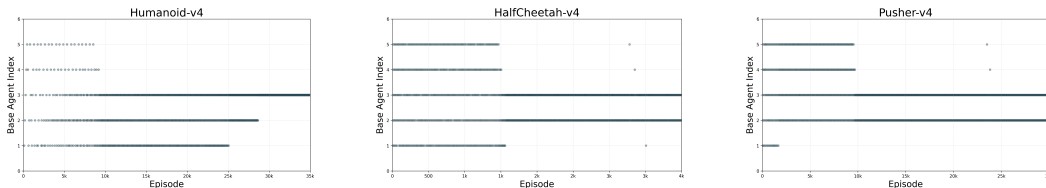

Figure 5: $D^3RB$ Selection Statistics in MuJoco environments

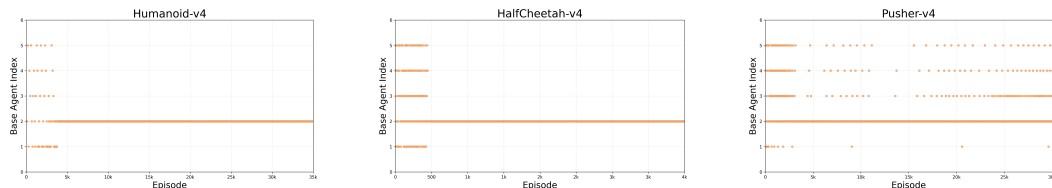

Figure 6: UCB Selection Statistics in MuJoco environments

The reward curves demonstrate that the choice of selector matters in the model selection task, as $D^3RB$ and $ED^2RB$ (which have similar model selection guarantees) achieve superior performance in comparison to other model selection strategies. Additionally, the reward curve of UCB highlights that

assumptions of Bandit learning might fail in RL model selection tasks. We observe that UCB is able to dominate other selectors at the initial phase of training, but the performance plateaus towards the end. We explain this behavior by investigating the detailed selection statistics of the model selection algorithm. Figure 5 illustrates how base agents are picked by selectors throughout the training. We observe that UCB is overcommitting to a base agent that was optimal at the initial stage of training, but fails to remain explorative and adapt to new step sizes that are optimal towards the end. We include the selection statistics of all algorithms in Appendix B.2. We further analyze the results of the Classic algorithm and the effect of misspecification test in Appendix A.5.

### 4.3 SELF-MODEL SELECTION

Self-model selection refers to the setting where base agents have identical configurations but are executed with different initial randomization (seed). This setting can stabilize the training dynamics of RL algorithms that are highly sensitive to the seeding. We show that by combining a number of base agents that each fail with a fixed probability, self-model selection can capitalize on the successful runs and perform as well as the best of them. The following theorem formalizes this property,

**Theorem 5.** *Let $\delta \in (0, 1)$, and $\mathcal{R}_\star(t, \delta) : ([T], (0, 1)) \to \mathbb{R}_+$ be a function satisfying*

$$\mathcal{R}_\star(t, \delta) \leq d_\star \sqrt{t} \qquad \forall t \in [T] \tag{13}$$

*Suppose a learning agent $\mathcal{B}$ satisfies,*

$$\mathbb{P}\left(\text{Regret}_t^{\mathcal{B}} \leq \mathcal{R}_\star(t, \delta) \quad \forall t \in [T]\right) \geq 1 - \gamma(\delta) \tag{14}$$

*for some $\gamma(\delta) \in (0, 1)$. Then, for $M = \lceil \frac{\log(\delta)}{\log(\gamma(\delta))} \rceil$, $D^3RB$ achieves the bound,*

$$\text{Regret}(T) = \mathcal{O}\left(d_* M \sqrt{T} + d_*^2 \sqrt{MT}\right)$$

*with probability at least $1 - \delta$.*

*Proof.* Appendix A.3

## 5 DISCUSSION AND FUTURE WORK

One can consider a similar model selection framework to ours, where base agents can communicate or share data. From the algorithmic perspective, a natural idea is to use importance sampling to update the policy of one base agent with the trajectory collected by another base agent. We discuss this in more detail in Appendix A.4, and leave the theoretical analysis and empirical gain of sharing data as an interesting future work.

The theoretical guarantee 11 of data-driven model selection reflects that the regret of the selector grows linearly with the number of base agents $M$. This is a barrier for deploying these methods to large-scale model selection tasks and incentivizes designing methods with improved dependency on $M$. Prior work of (Kassraie et al., 2024) has studied this problem for the case of linear bandits, but the question remains open for RL.

Apart from the model selection tasks studied in this paper, prior work has studied online reward selection (Zhang et al., 2024) in policy optimization methods. One can consider other practical model selection tasks, such as selection among agents with different exploration strategies or selection among different finetuning algorithms for language models.

## 6 CONCLUSION

This work presents a principled mechanism to improve the efficiency of RL training procedure via online model selection. We studied data-driven model selection methods, how they characterize misspecification of base agents, and their theoretical guarantees. Through careful theoretical analysis, we showed that these methods 1) learn to adaptively direct more compute towards the base agents with better realized performance, 2) adapt to new optimal choices under non-stationary dynamics, and 3) enhance training stability in presence of sensitivity to randomization. We validated these properties on several practical model selection tasks in deep RL, including neural architecture selection and step-size selection.

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

# APPENDIX

## TABLE OF CONTENTS

## A  THEORETICAL RESULTS

### A.1  MISSPECIFICATION TEST

The misspecification test determines whether the bound $\hat{d}_t^i \sqrt{n_t^i}$ matches the realized performance of the agent in a principled manner. For any base agents $j \in [M]$,

$$v^* \geq \frac{\bar{u}_t^j}{n_t^j} \qquad\qquad (\text{definition of } v^*)$$

$$\geq \frac{u_t^j}{n_t^j} - c\sqrt{\ln\frac{\frac{M \ln n_t^j}{\delta}}{n_t^j}} \qquad\qquad (\text{Event } \mathcal{E})$$

For a well-specified base agent $i \in [M]$ that satisfies its regret upper bound,

$$\text{Regret}_t^i = n_t^i v^* - \bar{u}_t^i \leq \hat{d}_t^i \sqrt{n_t^i}$$

Rearranging,

$$v^* \leq \frac{\bar{u}_t^i}{n_t^i} + \frac{\hat{d}_t^i \sqrt{n_t^i}}{n_t^i}$$

$$\leq \frac{u_t^i}{n_t^i} + c\sqrt{\ln\frac{\frac{M \ln n_t^j}{\delta}}{n_t^j}} + \frac{\hat{d}_t^i \sqrt{n_t^i}}{n_t^i} \qquad\qquad (\text{Event } \mathcal{E})$$

Therefore, a well-specified agent $i$ should satisfy,

$$\frac{u_t^i}{n_t^i} + c\sqrt{\ln\frac{\frac{M \ln n_t^i}{\delta}}{n_t^i}} + \frac{\hat{d}_t^i \sqrt{n_t^i}}{n_t^i} \geq \frac{u_t^j}{n_t^j} - c\sqrt{\ln\frac{\frac{M \ln n_t^j}{\delta}}{n_t^j}} \qquad \forall j \in [M]$$

and otherwise the agent is misspecified, resulting in test 12. Triggering this misspecification test implies that the bound $\hat{d}_t^i \sqrt{n_t^i}$ is too small to bound the realized regret of the agent and hence he algorithm doubles the estimated regret coefficient $\hat{d}_t^i$.

## A.2 PROOF OF THEOREM 4

We borrow the following lemmas from (Dann et al., 2024) to prove Theorem 4.

**Lemma 6.** *In event $\mathcal{E}$ 10, for each base agent $i \in [M]$ , the regret multiplier $\hat{d}_t^i$ in algorithm 1-left satisfies,*

$$\hat{d}_t^i \le 2d_t^i, \qquad \forall t \in \mathbb{N} \tag{15}$$

**Lemma 7.** *The potentials in Algorithm 1-(left) are balanced at all times up to a factor 3, that is for all $t \in [T]$,*

$$\phi_t^i \le 3\phi_t^j \qquad \forall i, j \in [M] \tag{16}$$

**Proof of Theorem 4** By Lemma 7, the potentials in algorithm satisfy,

$$\phi_t^i \le 3\phi_t^j \qquad \forall i, j \in [M] \tag{17}$$

Substitute the definition of $\phi_t^i$,

$$\hat{d}_t^i \sqrt{n_t^i} \le 3\hat{d}_t^j \sqrt{n_t^j} \Rightarrow (\hat{d}_t^i)^2 n_t^i \le 9(\hat{d}_t^j)^2 n_t^j \tag{18}$$

Rearrange,

$$n_t^i \le 9 \frac{(\hat{d}_t^j)^2}{(\hat{d}_t^i)^2} n_t^j \tag{19}$$

By lemma 6, we have $\hat{d}_t^i \le 2d_t^i$,

$$\le 9 \frac{(d_t^j)^2}{(d_t^i)^2} n_t^j = 9 \frac{(1/d_t^i)^2}{(1/d_t^j)^2} n_t^j \tag{20}$$

For all $t \in [T]$ and a fixed $j \in [M]$,

$$t = \sum_{i=1}^{M} n_t^i = n_t^j \left( \sum_{i=1}^{M} \frac{(1/d_t^i)^2}{(1/d_t^j)^2} \right) = \frac{n_t^j}{(1/d_t^j)^2} \sum_{i=1}^{M} (1/d_t^i)^2 \tag{21}$$

Rearranging yields the final result in theorem 4,

$$n_t^j = \frac{(1/d_t^j)^2}{\sum_{i=1}^{M} (1/d_t^i)^2} \, t \tag{22}$$

## A.3 PROOF OF THEOREM 5

**Proof of Theorem 5**: Suppose agent $\mathcal{B}$ satisfies its theoretical upper bound with probability at least $1 - \gamma(\delta)$,

$$\mathbb{P} \left( \text{Regret}_t^{\mathcal{B}} \le \mathcal{R}_\star(t, \delta) \quad \forall t \in [T] \right) \ge 1 - \gamma(\delta) \tag{23}$$

Then, if we combine $M$ independent base agents of type $\mathcal{B}$, the probability of at least one of them succeeding is larger than $1 - \gamma(\delta)^M$. Therefore, self-model selection requires

$$1 - \gamma(\delta)^M = 1 - \delta \Rightarrow M = \lceil \frac{\log(\delta)}{\log(\gamma(\delta))} \rceil \tag{24}$$

base agents to achieve its bound with probability $1 - \delta$.

## A.4 Sharing data via Importance sampling

**Theorem 8.** *Consider the case of episodic RL, where we roll out policy $\pi^i$, and collect a trajectory $\tau = (s_1, a_1, r_1, \ldots, s_T, a_T, r_T) \sim \mathbb{P}^i(\tau)$. Here, $\mathbb{P}^i(\tau)$ is the joint distribution over the trajectory $\tau$ under policy $\pi^i$,*

$$\mathbb{P}^i(\tau) = \mathcal{D}(s_1)\pi^i(a_1|s_1)\mathcal{R}(r_1|s_1,a_1)\mathcal{P}(s_2|s_1,a_1)\ldots\pi^i(a_T|s_T)\mathcal{R}(r_T|s_T,a_T)$$

*Then,*

$$\frac{\mathbb{P}^j(\tau)}{\mathbb{P}^i(\tau)} \sum_{t=1}^{T} \gamma^t r_t$$

*is an unbiased estimate of the discounted episodic reward that we would've collected under $\pi^j$.*

*Proof.* By definition,

$$\mathcal{J}(\pi^i) = \mathbb{E}\left[\sum_{t=1}^{T} \gamma^{t-1} r_t \mid \tau \sim \mathbb{P}^i\right] = \mathbb{E}_{\tau \sim \mathbb{P}^i}\left[\sum_{t=1}^{T} \gamma^{t-1} r_t\right]$$

By importance sampling,

$$\mathbb{E}_{\tau \sim \mathbb{P}^i}\left[\frac{\mathbb{P}^j(\tau)}{\mathbb{P}^i(\tau)} \sum_{t=1}^{T} \gamma^t r_t\right]$$

$$= \sum_{\tau} \left(\frac{\mathbb{P}^j(\tau)}{\mathbb{P}^i(\tau)} \sum_{t=1}^{T} \gamma^t r_t\right) \mathbb{P}^i(\tau) = \sum_{\tau} \left(\mathbb{P}^j(\tau) \sum_{t=1}^{T} \gamma^t r_t\right)$$

$$= \mathbb{E}_{\tau \sim \mathbb{P}^j}\left[\sum_{t=1}^{T} \gamma^t r_t\right] = \mathbb{E}\left[\sum_{t=1}^{T} \gamma^{t-1} r_t \mid \tau \sim \mathbb{P}^j\right] = \mathcal{J}(\pi^j)$$

**Proposition 9.** *For a given trajectory $\tau = \{(s_t, a_t, r_t)\}_{t=1}^{T}$ collected by policy $\pi^i$,*

$$(\Pi_t \alpha_t) \sum_{t=1}^{T} \gamma^t r_t, \quad \text{where} \quad \alpha_t = \frac{\pi^j(a_t|s_t)}{\pi^i(a_t|s_t)}$$

*is the importance sampling ratio for the discounted episodic reward that we would have collected under policy $\pi^j$.*

*Proof.*

$$\mathbb{E}_{\tau \sim \mathbb{P}^i}\left[\frac{\mathbb{P}^j(\tau)}{\mathbb{P}^i(\tau)} \sum_{t=1}^{T} \gamma^t r_t\right]$$

$$= \mathbb{E}_{\tau \sim \mathbb{P}^i}\left[\frac{\mathcal{D}(s_1)\pi^j(a_1|s_1)\mathcal{R}(r_1|s_1,a_1)\mathcal{P}(s_2|s_1,a_1)\ldots\pi^j(a_T|s_T)\mathcal{R}(r_T|s_T,a_T)}{\mathcal{D}(s_1)\pi^i(a_1|s_1)\mathcal{R}(r_1|s_1,a_1)\mathcal{P}(s_2|s_1,a_1)\ldots\pi^i(a_T|s_T)\mathcal{R}(r_T|s_T,a_T)} \sum_{t=1}^{T} \gamma^t r_t\right]$$

$$= \mathbb{E}_{\tau \sim \mathbb{P}^i}\left[\frac{\pi^j(a_1|s_1)\ldots\pi^j(a_T|s_T)}{\pi^i(a_1|s_1)\ldots\pi^i(a_T|s_T)} \sum_{t=1}^{T} \gamma^t r_t\right] = \mathbb{E}_{\tau \sim \mathbb{P}^i}\left[(\Pi_t \alpha_t) \sum_{t=1}^{T} \gamma^t r_t\right]$$

## A.5 Misspecification Test based on Realized versus Expected performance

We consider the role of the misspecification test in the performance of model selection algorithms. We analyze and compare two of the algorithms, D³RB and Classic Balancing, that select base agents by performing a misspecification test.

Recall the set of base agents, $\mathcal{B} = \{\mathcal{B}^1, \ldots, \mathcal{B}^M\}$ and suppose $\mathcal{B}^i$ has a high-probability upper bound $\mathcal{R}^i(t, \delta)$,

$$\{\mathcal{R}^1(t, \delta), \ldots, \mathcal{R}^M(t, \delta)\}$$

Suppose $\mathcal{B}^i$ is an instance of a sequential decision-making algorithm with high-probability regret guarantee $\mathcal{R}^i(t, \delta)$.

$$\mathbb{P}\left[\text{Regret}_t^i \leq \mathcal{R}^i(t, \delta)\right] \geq 1 - \delta \tag{25}$$

Using these theoretical regret bounds, we can perform a similar misspecification test to 12,

$$\frac{u_t^i}{n_t^i} + c\sqrt{\ln \frac{M \ln n_t^i}{\delta}} + \frac{\mathcal{R}^i(t, \delta)}{n_t^i} \leq \max_{j \in [M]} \frac{u_t^j}{n_t^j} - c\sqrt{\ln \frac{M \ln n_t^j}{\delta}} \tag{26}$$

At round $t$, the Classic Balancing algorithm selects the base agent with minimum regret bound $i_t = \arg\min_{i \in [M]} R(t, \delta)$, eliminating the base agents that are flagged as misspecified by test 26. Note that, the difference between tests 12 and 26 is that one is performed based on the theoretical bound, and the other is performed using the data-adaptive regret bound actively estimated by realized rewards. The drastic difference between the performance of $D^3RB$ and Classic Balancing selectors in figure 4 highlights the role of designing data-adaptive model selection algorithms versus selectors that perform upon theoretical bounds on the expected performance of the agent.

## B  EXPERIMENTAL DETAILS

The following is the detailed architectural choice of three base agents in experiment 4.1. The second architecture has the configuration with best realized performance. The first architecture, roughly shares the same number of parameters, but only has one convolutional layer, limiting the agents ability in tasks that require temporal reasoning. The third architecture shares that same number of layers to the second one, but has less parameters that limits the representational capacity of the agent.

### B.1  NEURAL ARCHITECTURE SELECTION

$\mathcal{B}^1$ network architecture:

```
1    network = nn.Sequential(
2            nn.Conv2d(4, 16, 8, stride=4),
3            nn.ReLU(),
4            nn.Flatten(),
5            nn.Linear(6400,256),
6            nn.ReLU(),
7            nn.Linear(256, env.single_action_space.n),
8        )
```

$\mathcal{B}^2$ network architecture:

```
1    network = nn.Sequential(
2            nn.Conv2d(4, 32, 8, stride=4),
3            nn.ReLU(),
4            nn.Conv2d(32, 64, 4, stride=2),
5            nn.ReLU(),
6            nn.Conv2d(64, 64, 3, stride=1),
7            nn.ReLU(),
8            nn.Flatten(),
9            nn.Linear(3136, 512),
10           nn.ReLU(),
11           nn.Linear(512, env.single_action_space.n),
12       )
```

$\mathcal{B}^3$ network architecture:

```
1    network = nn.Sequential(
2            nn.Conv2d(4, 8, 8, stride=4),
3            nn.ReLU(),
4            nn.Conv2d(8, 8, 4, stride=2),
5            nn.ReLU(),
6            nn.Conv2d(8, 8, 3, stride=1),
7            nn.ReLU(),
8            nn.Flatten(),
9            nn.Linear(392, 32),
10           nn.ReLU(),
11           nn.Linear(32, env.single_action_space.n),
12       )
```

## B.2 STEP SIZE SELECTION

Here, we include the selection statistics plot for the rest of algorithms in experiment B.2.

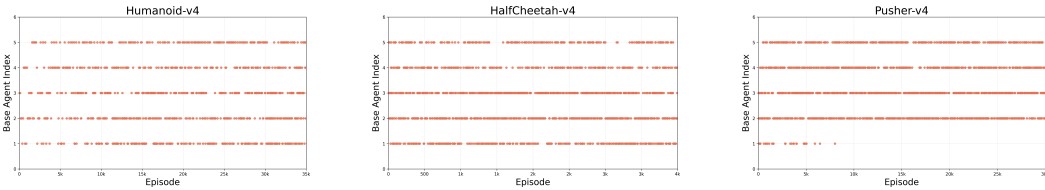

Figure 7: Exp3 Selection Statistics

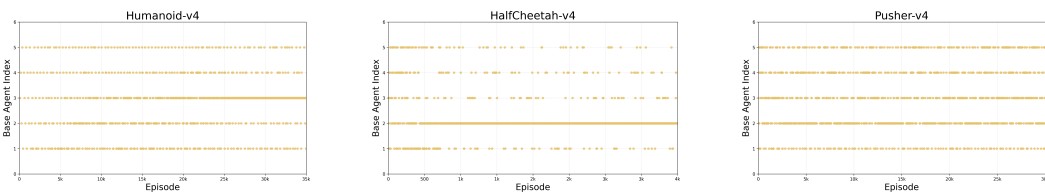

Figure 8: Corral Selection Statistics

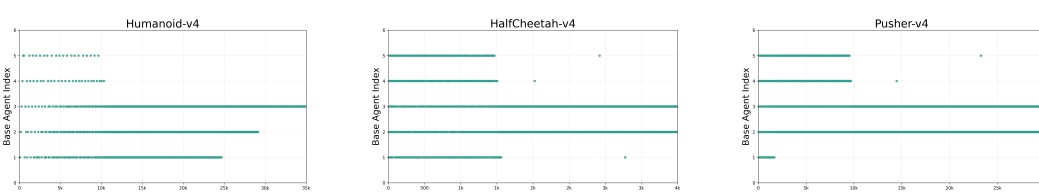

Figure 9: ED$^2$RB Selection Statistics

# C PSEUDO-CODES OF MODEL SELECTION ALGORITHMS

## C.1 MODEL SELECTION ALGORITHMS

In this section, we provide 5 model selection strategies that follow the interface in algorithm 1-left. To avoid including all the theoretical details, there might be a slight abuse of notation in the pseudocodes.

### ED$^2$RB

Estimating Data Driven Regret Balancing (ED$^2$RB) (Dann et al., 2024) is similar to D3RB, though it tries to directly estimate the regret coefficients.

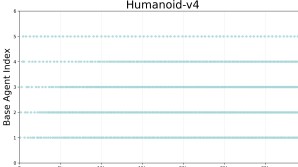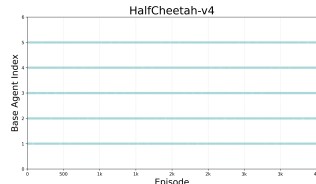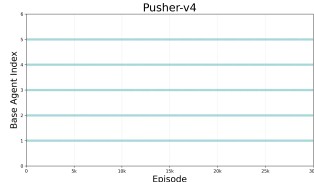

Figure 10: Classic Selection Statistics

---

**Algorithm 3** ED$^2$RB

---

**Input:** $m$, $\beta$, $\Psi$, $\delta$
**Function** `sample()`:
    // Sample base index
    $i = \arg\min_j \Psi_j$   $\pi_i, \alpha_i \leftarrow \beta_i$   **return** $i, \pi_i, \alpha_i$

**Function** `update(`$i$, $R[1:T]$`)`:
    $R_{norm} \leftarrow normalize(R[1:T])$   // Update Statistics
    $u^i = u^i + R_{norm}$   $n^i = n^i + 1$   // Estimate active regret coefficient
    $d^i = \max\{d_{min}, \sqrt{n_t^{i_t}}(max_j \frac{u^j}{n^j} - c\sqrt{ln\frac{Mlnn^j}{\delta}}{n^j} - c\sqrt{ln\frac{mlnn^i}{\delta}}{n^i} - \frac{u^i}{n^i})\}$   // Update balancing potential
    $\Psi^i = clip(d^i\sqrt{n^i}, \Psi^i, 2\Psi^i)$

---

CLASSIC BALANCING

The Classic Regret Balancing Algorithm (Pacchiano et al., 2020a) starts with the full set of base agents $\beta = [\beta_1, ..., \beta_m]$, at each round, the algorithm performs miss-specification on each of the base agents and eliminates the miss-specified one. Denote $\Psi^j$ as the empirical regret upper bound of base agent $j$.

---

**Algorithm 4** Classic Balancing

---

**Input:** $m$, $\beta$, $\Psi$, $\delta$
**Function** `sample()`:
    // Sample Base index
    $i = \arg\min_j \Psi_j$   $\pi_i, \alpha_i \leftarrow \beta_i$   **return** $i, \pi_i, \alpha_i$

**Function** `update(`$i$, $R[1:T]$`)`:
    $R_{norm} \leftarrow normalize(R[1:T])$
    // Update statistics
    $u^i = u^i + R_{norm}$   $n^i = n^i + 1$
    // Perform miss-specification test for all the remaining base agents
    **for** $\beta_k \in \beta$ **do**
        $\frac{u^k}{n^k} + \frac{d^k\sqrt{n^k}}{n^i} + c\sqrt{ln\frac{mlnn^k}{\delta}}{n^k} \leq max_j\frac{u^j}{n^j} - c\sqrt{ln\frac{Mlnn^j}{\delta}}{n^j}$   **if** *miss-specified* **then**
            $\beta \leftarrow \beta/\{\beta_k\}$

---

EXP3

Exponential-weight algorithm for exploration and exploitation (EXP3) learns a probability distribution $\Psi^i = \frac{exp(S^i)}{\sum_{j=1}^m exp(S^j)}$ over base learners, where $S^i$ is a total estimated reward of base agent $i$ up to this round.

---

**Algorithm 5** EXP3

---

**Input:** $m$, $\beta$, $\Psi$, $\delta$

  **Function** `sample()`:

    | // Sample Base index
    | $i = \arg\max_j \Psi_j$
    | $\pi_i, \alpha_i \leftarrow \beta_i$    **return** $i, \pi_i, \alpha_i$

  **Function** `update`$(i, R[1:T])$:

    | $R_{norm} \leftarrow normalize(R[1:T])$  // Update statistics
    | **for** $j \in 1, ..., m$ **do**
        | $S^j = S^j + 1 - \frac{\mathbb{I}\{j=i\}(1-R_{norm})}{\Psi^i}$
    | // Update Distribution
    | $\Psi^i = \frac{exp(S^i)}{\sum_{j=1}^m exp(S^j)}$

---

### CORRAL

Corral (Agarwal et al., 2017) learns a distribution $\Psi$ over base agents and updates it according to LOG-BARRIER-OMD algorithm. We skip the algorithmic details and refer to the updating rule mentioned in the original paper as Corral-Update.

---

**Algorithm 6** Corral

---

**Input:** $m$, $\beta$, $\Psi$

  **Function** `sample()`:

    | // Sample base index
    | $i \sim \Psi$
    | $\pi_i, \alpha_i \leftarrow \beta_i$    **return** $i, \pi_i, \alpha_i$

  **Function** `update`$(i, R[1:T])$:

    | $R_{norm} \leftarrow normalize(R[1:T])$  // Update according to Corral
    | $\Psi^j \leftarrow$ Corral-Update$(R_{norm})$

---

### UCB

The Upper Confidence Bound algorithm (UCB) maintains an optimistic estimate of the mean for each arm (Lattimore and Szepesvári, 2020). Denote $\Psi^i$ as the upper confidence bound of arm $i$. The UCB algorithm for learning rate-free RL works as follows,

---

**Algorithm 7** UCB

---

**Input:** $m$, $\beta$, $\Psi$, $\delta$

**Function** `sample()`:

    |   // Sample base index
    | $i = \arg\max_j \Psi_j$
    | $\pi_i, \alpha_i \leftarrow \beta_i$    **return** $i, \pi_i, \alpha_i$

**Function** `update`$(i, R[1:T])$:

    | $R_{norm} \leftarrow normalize(R[1:T])$  // Update statistics
    | $u^i = u^i + R_{norm}$   $n^i = n^i + 1$   $\mu^i = \frac{u^i}{n^i}$
    | // Update Upper Confidence Bounds
    | $\Psi^i = UCB^i(\delta) = \mu^i + \sqrt{\frac{2log(1/\delta)}{n^i}}$

---

### C.2 RL ALGORITHMS

Two of the predominant approaches for learning the (near) optimal policy in reinforcement learning are policy optimization and Q-learning. Policy optimization starts with an initial policy and in each

episode updates the parameters by taking gradient steps toward maximizing the episodic return. Denote learning rate as $\alpha \in \mathbb{R}$, a common update rule in policy optimization methods is

$$\theta \leftarrow \theta + \alpha \, \mathbb{E}\left[\sum_{t=0}^{T} \nabla_\theta \log \pi_\theta(s_t, a_t)(Q^{\pi_\theta}(s_t, a_t) - V^{\pi_\theta}(s_t))\right] \tag{27}$$

Q-learning uses the temporal differences method to update the parameters of $Q^{\pi_\theta}$. A common update rule is

$$\theta \leftarrow \theta + \alpha \, \mathbb{E}_{s,a,s',r \sim D}\left[\nabla_\theta (r + \gamma \max_{a' \in A} Q^{\pi_{\bar{\theta}}}(s', a') - Q^{\pi_\theta}(s, a))^2\right] \tag{28}$$

where $D$ is the experience replay buffer and $\bar{\theta}$ is a frozen parameter set named target parameter. Proximal Policy Optimization (PPO) (Schulman et al., 2017) and Deep Q-Networks (DQN) (Mnih et al., 2015) follow the first and second approaches, respectively.

