# OpenReview forum: "Select the Right Agent: Data-Driven Online Model Selection in Reinforcement Learning"
_ICLR.cc/2026/Conference — Submitted to ICLR 2026_

### Official Review · Reviewer_VTaj · 2025-10-26

**Soundness:** 4
**Presentation:** 4
**Contribution:** 4
**Rating:** 8
**Confidence:** 3

**Summary:**

The idea is to use a selector meta algorithm that adaptively choose from a set of base RL agents during a training run.

**Strengths:**

* it shows strong empirical valiadtion in section 4
*  The paper is well-written and easy to follow.
* The paper does an excellent job of connecting theory to practice. Theorem 4 provides a clean, intuitive theoretical justification for why the the selector works: it adaptively allocates more "pulls" (training episodes) to agents that demonstrate a better (lower) regret coefficient.

**Weaknesses:**

* the authros didn't provide the evidence that the model is scalable
*  The paper relies heavily on D³RB but didn't express clearly why not ED²RB

**Questions:**

n/a

---

### Official Review · Reviewer_MRCK · 2025-10-30

**Soundness:** 2
**Presentation:** 2
**Contribution:** 2
**Rating:** 2
**Confidence:** 3

**Summary:**

This paper studies online model selection in RL. It integrates a data-driven model selction method D3RB into standard RL loops. The paper shows the efficiency compute allocation, adaptation to non-stationary envs, and improved stability through a mix of theoretical restatements and small-scale experiments on Atari and MuJoCo tasks.

**Strengths:**

Clear motivation, and the targeted problem, RL algos often require many hyperparameters and are sensitive to misspecification,  is critical.

**Weaknesses:**

1. Lack of novelty. The primary algorithmic component (D3RB) and its theoretical guarantees are directly borrowed from previous works.
2. Even the central theorems are re-statements of known results. There is no new analysis specific to RL, such as, How model selection interacts with stochastic policy gradients? Whether the selector biases learning or affects convergence? Theoretical treatment of non-stationary dynamics or exploration coupling?
3. Related work discussion omits recent AutoRL and hyperparameter optimization literature.

**Questions:**

1. In the main paper, the theorems are derived under the episodic RL setting without discount factor $\gamma$, but in Appendix A.4, it uses discount factor, here exists mathematically inconsistency.
2. Experiments use only three random seeds which is unacceptable for RL where variance is large.
3. No numerical variance, confidence intervals, or significance tests are provided. The error bars on Base Agent 2 are nearly invisible.
4. The paper claims “better resource allocation” but provides no runtime, compute usage, or sample efficiency data.
5. The authors repeatedly claim to “improve efficiency” and “stabilize training,” but the evidence only shows that D3RB eventually matches the best base agent. There is no demonstration of faster convergence or less compute spent for the same return.
6. Can D3RB outperform all base agents by better combining partial learning progress, or does it only track the best one?

---

### Official Review · Reviewer_BbGC · 2025-11-01

**Soundness:** 3
**Presentation:** 3
**Contribution:** 2
**Rating:** 4
**Confidence:** 3

**Summary:**

The paper studies online model selection for RL by embedding D$^3$RB (or any model selection algorithm following similar interface to D$^3$RB) directly into the RL training loop.
Concretely, the proposed method adopts D$^3$RB’s balancing potential $\phi\_t^i=\hat{d}\_t^i\sqrt{n\_t^i}$, regret and regret coefficient, and the misspecification test, and then provides a minimal Selector+RL integration that replaces the (contextual) bandit value definitions with episodic RL returns.
Theoretical guarantees are inherited from D3RB (i.e., the selector’s high-probability regret bound is adopted rather than strengthened).
Beyond this transfer, this paper analyzes how compute is allocated across agents within RL and proves a relationship between the allocated compute and regret coefficient of each base learner (i.e., $\alpha_i\propto 1/d\_i^2$) clarifying why better-performing agents get more resource.
Overall, relative to D³RB, the main contribution is an RL-oriented integration and interpretation of the selector’s behavior in episodic settings.

Empirically, the paper evaluates three tasks: (1) neural architecture selection task in DQN agents for four Atari games; (2) step-size selection task in RL (especially, PPO agents in three different MuJoCo environments); and (3) self-model selection across different random seeds.
These case studies support the claims that data-driven selection adapts under non-stationarity and stabilizes training without requiring agent-specific structure or regret bounds.

**Strengths:**

- Proposes applying D$^3$RB as a selector within the RL training loop and clearly describes a minimal integration recipe that reuses D$^3$RB’s interface (balancing potential, misspecification test, etc.), which makes the approach easy to drop into existing pipelines.
- Provides empirical comparisons across multiple selector baselines (e.g., bandit-style selectors), illustrating adaptation under non-stationarity and improved training stability.
- Does not require agent-specific structure or per-agent regret bounds, which increases applicability across RL algorithms.
- Offers an interpretive analysis of compute allocation and proves a proportionality result $\alpha_i\propto 1/d\_i^2$, giving a useful explanation for why stronger agents receive more updates.

**Weaknesses:**

- Theoretical novelty beyond D$^3$RB is minimal. Core guarantees are inherited without non-stationary (e.g., switching-regret) results.
- Evaluation scope is narrow (few Atari/MuJoCo tasks, limited seeds), making generalization claims tentative.
- Ablations (e.g., sensitivity to $d\_{\min}$​, selector variants like ED$^3$RB) and baselines (e.g., meta learning algorithms for adaptive hyperparameter selection in RL, mentioned in line 65) are thin, limiting insight into when the approach helps or fails.

**Questions:**

- In Fig. 3, why does the Selector (D³RB) performance curve differ from any mixture/combination of the base agents’ learning curves, even though the selector is claimed not to affect each base algorithm’s training curve?
- Why do Fig. 3 and 4 use different environments, even though it seems feasible to include the same baselines in Fig. 3 and to report the base agents’ standalone performance in Fig. 4? Could you harmonize the setups so that (1) Fig. 3 includes the baselines shown in Fig. 4 and (2) Fig. 4 also plots the standalone base-agent curves?

---

### Official Review · Reviewer_9SkD · 2025-11-05

**Soundness:** 3
**Presentation:** 3
**Contribution:** 2
**Rating:** 4
**Confidence:** 3

**Summary:**

This paper applies online model-selection to deep reinforcement learning by using D³RB (Dann et al. 2024) to adaptively select between different RL agents (with different architectures, hyperparameters, or seeds) to update after each episode. The goal is to allocate resources to promising agents, improving stability and avoiding committing to suboptimal configurations. The method is evaluated on Atari (DQN) and MuJoCo (PPO) and generally matches or improves the performance of the best agent in hindsight.

**Strengths:**

Achieving stability across seeds and hyperparameters remains a real issue in deep RL practice. Wrapping a model-selection algorithm around existing RL agents is appealing and broadly applicable. The D^3RB is a provably-sound approach that is shown to improve training robustness across a range of settings (architecture choice, learning rate, seed selection) in deep RL contexts where algorithm selection is typically heuristic.

**Weaknesses:**

The theoretical novelty is limited/unclear. Section 3 restates prior results from Dann et al. (2024), and while Section 4 adds some analysis, the contribution beyond prior theory is not fully clear. For instance, Theorem 5 appears as a simple corollary, and Theorem 4 appears more as a sanity-check. This makes it difficult to appreciate the paper since the positioning is primarily theoretical.

On the experimental side, the results are promising and I feel that, if anything, the paper should be positioned as more of an experimental paper rather than a theoretical paper as it currently stands. However, if it was actually an experimental paper, one would expect the experimental scope to be broader. While Atari and MuJoCo are good starting points, the real impact of model selection should be settings where we could not simply afford to just run the M agents in parallel. How would these algorithms fare e.g. in LLM post-training settings, where compute is comparatively more valuable. Regarding the baselines, the authors mainly compare to theoretical model-selection algorithms and do not consider more empirical baselines for hyperparameter-tuning (e.g. hyperband/meta-learning algorithms), which would significantly strengthen the case.  Lastly, the compute overhead of the model-selection algorithm (while it appears at a distance to be somewhat minimal) should be tracked.

Overall, it appears the work sits between theory and practice, but the contributions on the theoretical side are somewhat minimal. If framed primarily as a practical contribution, it needs deeper empirical analysis; if framed as theory, the novelty is less clear. Still, the contribution is a promising an empirical demonstration that D³RB can be usefully integrated into deep RL pipelines.

**Questions:**

1.	Can you clarify the precise theoretical novelty beyond Dann et al. (2024)?
	2.	Is there a reason not compare against more empirical tuning strategies such as meta-learning?

---

### Meta-Review · Area_Chair_4NRM · 2025-12-09

**Summary:**

The paper addresses online model selection in reinforcement learning (RL) and proposes a data-driven method to enhance training efficiency and performance. It focuses on adaptive selection of RL agents using a theoretical framework that emphasizes efficient resource allocation, stabilized training, and adaptation to non-stationary dynamics. Accompanying theoretical developments are empirical results demonstrating the effectiveness of the proposed approach on various RL tasks.

Reviewers acknowledge the clear motivations behind the work, noting the RL algorithms are particularly sensitive to hyperparameter selection. However, they all voice serious concerns regarding the paper's contributions. They note that the proposed method derivative, arguing that the main theoretical guarantees are provided by D3RB and thus raising questions about novelty. Reviewers additionally point out that the theoretical analysis does not sufficiently address interactions between model selection and crucial RL dynamics, and that the empirical evaluation is limited both in scope and depth, and thus the paper does not convincingly demonstrate scalability or comprehensively address practical challenges. Overall, the contributions appear to straddle the line between theoretical insight and practical application, with neither being satisfactorily developed. The authors did not provide a rebuttal and the recommendation is thus to Reject.

**Reviewer Concerns:**

The authors provided no rebuttal.

**Reviewer Scores:**

The authors provided no rebuttal, but I do not think the reviewers would have been convinced to raise their scores.

---

### Decision · Program_Chairs · 2026-01-26

Reject